# Clinical Characteristics and Underlying Factors Related to Recovery from Severe Pneumonia in Under-Five Children with or without Malnutrition Treated at Health Care Facilities in Bangladesh

**DOI:** 10.3390/children8090778

**Published:** 2021-09-04

**Authors:** K. A. T. M. Ehsanul Huq, Michiko Moriyama, Ryota Matsuyama, Mohammod Jobayer Chisti, Md Moshiur Rahman, Nur Haque Alam

**Affiliations:** 1Graduate School of Biomedical and Health Sciences, Hiroshima University, Kasumi 1-2-3, Minami-ku, Hiroshima 734-8553, Japan; morimich@hiroshima-u.ac.jp (M.M.); rmatsuyama@hiroshima-u.ac.jp (R.M.); moshiur@hiroshima-u.ac.jp (M.M.R.); 2International Centre for Diarrhoeal Disease Research (icddr,b), Dhaka 1212, Bangladesh; chisti@icddrb.org (M.J.C.); nhalam@icddrb.org (N.H.A.)

**Keywords:** clinical characteristics, severe pneumonia, under-five children, malnutrition, Bangladesh

## Abstract

Severe pneumonia with co-morbidity of malnutrition is one of the leading causes of death among children younger than five years of age. We aimed to compare the clinical features related to recovery from severe pneumonia in malnourished and well-nourished under-five children. A significantly lesser proportion of malnourished children compared to well-nourished recovered from rapid breathing (86.5% vs. 90.2%; *p* = 0.035), chest wall in-drawing (90.5% vs. 93.9%; *p* = 0.019), and fever (92.2% vs. 95.2%; *p* = 0.021) on day six after admission to health-care facilities. Malnourished children with severe pneumonia had significantly more rapid breathing (Adjusted Odds Ratio (AOR)) 1.636, 95% Confidence interval (CI) 1.150–2.328 *p* = 0.006), chest wall in-drawing (AOR 1.698, 95% CI 1.113–2.590, *p* = 0.014), and fever (AOR 1.700, 95% CI 1.066–2.710, *p* = 0.026) compared to those in well-nourished children. The study results suggested the lesser disappearance of clinical features among the malnourished children in Bangladesh underscores their vulnerability to severe pneumonia.

## 1. Introduction

Globally, pneumonia is the foremost infectious cause of death in children younger than five years of age, and malnourished children are more vulnerable to pneumonia [1]. Pneumonia and malnutrition are independently related to excess child deaths, and the combination of pneumonia and malnutrition has an additive adverse effect on child morbidity and deaths [2]. It has been reported that severely underweight (<3 weight-for-age Z-score; ZWA) and moderately wasted (>3 and <2 weight-for-height Z-score; ZWH) children in different developing countries having pneumonia were at 6.4- and 4.2-times higher risk of mortality, respectively, compared to the children without pneumonia [3]. On the other hand, children with the co-morbidity of pneumonia and severe malnutrition were found to be at a 15 times higher risk of deaths compared to those without severe malnutrition [3,4]. Moreover, malnutrition without any co-morbidity has been reported to be associated with eight times higher case fatality compared to those without malnutrition [5]. Worldwide, 21.9%, 7.3%, and 2.4% of under-five children were stunted, wasted, and severely wasted, respectively, in 2018 [6], and 45% of under-five deaths are related to malnutrition—mostly in low- and middle-income countries [7]. While the death of malnourished children has been on the decline over the last few decades, the rate is still high [8]. A significant effort needs to be made to reach the Sustainable Development Goal (SDG) of reducing deaths to less than 25 per 1000 live births by the year 2030 [9]. In Bangladesh, 13% of under-five childhood deaths occurred due to pneumonia [10], and 30.8% and 8.4% were stunted and wasting, respectively, in 2018 [11]. The indicators of undernutrition, namely stunting, wasting, and underweight will be reduced from 55.0% to 28.8%, 15.9% to 12.3%, and 61.8% to 17.4%, respectively, between 1990 and 2030 [12], yet the rates are much higher than the global target set for the SDGs of 15% stunting and 6% wasting among under-five children by the year 2030 [13].

An early study observed that rapid breathing and lower chest wall in-drawing were the best clinical predictors for hypoxemia and fatal morbidities related to severe pneumonia among severely malnourished under-five children [14]. Another study found that weight-for-age (WAZ) less than −2 to −3 SD, rapid breathing (≥ 70 breaths/min), and fever (axillary temperature ≥ 39 °C) were the risk factors for childhood pneumonia-related death [15].

Appropriate interventions are essential to prevent and treat malnourished under-five children with severe pneumonia to enhance child survival efforts [16]. As a significant proportion of children with severe pneumonia in developing countries have underlying malnutrition of various degrees [17,18], the systematic management of both problems is essential to prevent deaths [19]. The World Health Organization (WHO) developed a guideline for the management of childhood pneumonia at primary health care facilities and referral of those with severe pneumonia to a higher-level health care facility or a hospital [20]. Primary health care facilities are designed to manage children with non-severe pneumonia by paramedics and are not equipped to manage severe pneumonia [21]. In hospitals, severe pneumonia children are managed with parenteral antibiotics, oxygen therapy, supportive care, i.e., antipyretic for fever, bronchodilator for wheezing, suction for removal of thick secretion, maintaining fluid and nutritional balance, and careful monitoring [22].

Earlier efficacy trials have reported primary health care facilities equipped with oxygen, pulse oximeter, suction machine, and nebulizer facilities can provide efficient care for severe pneumonia irrespective of the nutritional status of the children on a day-care basis if staff were provided with quality clinical management training [23,24,25]. Such facilities provided care and treatment to the children on an outpatient basis, during daytime hours, and sent the children home with advice to return the following day until an outcome (recovery or referral to the hospital) was achieved [23,24,25].

Within the above-mentioned scenario, this study aimed to compare the resolution of clinical features of severe pneumonia among the malnourished and well-nourished under-five-year-old children at day-care and hospital settings on day six after admission, using indicators such as no increased (respiratory rate) breathing, no lower chest wall in-drawing, no hypoxemia, no nasal discharge, and no fever, and sociodemographic factors. 

## 2. Materials and Methods

### 2.1. Study Population

This was an observational study that compared malnourished and well-nourished children using the data from the same cohort (ClinicalTrials.gov Identifier: NCT02669654). The study was conducted in urban Dhaka, Bangladesh, between January 2016 and January 2019. We screened and enrolled children of both sexes aged 2 to 59 months, living in the study areas with severe pneumonia, with parent/caregiver-provided written informed consent administered by study nurses as inclusion criteria. After attending a health care facility, at first, study nurses checked the inclusion and exclusion criteria. We excluded children having severe acute malnutrition (SAM, defined as a weight-for-height Z-score (ZWH) < 3, or bilateral pitting edema, or a mid-upper arm circumference (MUAC) of <115 mm), occurring alone or in combination, due to their need for special management (e.g., nasogastric tube feeding, micronutrients, etc.). We combined the data of all children diagnosed with severe pneumonia who were admitted to the day-care or hospital care and grouped them on the basis of nutritional status, as with-malnutrition (defined as Z-score < −2 (weight-for-age; ZWA)) and without-malnutrition (defined as Z-score ≥ −2 (weight-for-age; ZWA)) [26]. We compared the baseline characteristics, resolution of clinical features of severe pneumonia, and other admission factors between the children with and without malnutrition. 

### 2.2. Study Areas

This study was conducted within the existing health systems of Bangladesh for the recruitment of study participants. In Dhaka city, the capital of Bangladesh, primary health care services are delivered directly by the Ministry of Local Government of Bangladesh (LGRD) and partly by the local non-government organizations (NGOs) supported by the United States Agency for International Development (USAID) and the Department for International Development (DFID). We selected 16 wards run by the local NGOs (Population Services & Training Center (PSTC), Bangladesh; Concerned Women for Family Development (CWFD), Bangladesh). Wards are the smallest local government units to provide primary health care, with populations between 50,000 and 80,000. The primary health care service of each ward is covered by one Primary Health Care Centre (PHCC). We recruited study children from those attending 16 PHCCs at 16 different ‘wards’ of the city. Of the 16 wards, 8 PHCCs from 8 wards treated pneumonia children on an outpatient basis and established day-care facilities at PHCCs for treating severe pneumonia children. We set up two day-care facilities in two PHCCs, located conveniently (based on distance) to cover the treatment of the population of 8 wards, i.e., for each to cover four wards. The children of the remaining 8 wards visiting the 8 PHCCs for the treatment of pneumonia and severe pneumonia children were referred to the nearby hospitals and clinics. We purposively selected the PHCCs and hospitals on the basis of infrastructure (adequate space, water supply, electricity, and willingness to perform the study).

### 2.3. Ethical Clearance

Ethical approval (PR#20124) was obtained from the Research Review Committee (RRC) and Ethical Review Committee (ERC) of the International Centre for Diarrhoeal Disease Research, Bangladesh (icddr,b) on 2 January 2021 and 18 February 2021, respectively. 

### 2.4. Study Procedures

Community Service Providers (CSPs) who resided in the community identified the under-five children in their catchment areas. Children with suspected pneumonia, with or without malnutrition, were brought by the Community Health Workers (CHWs) from their home or they were self-referred by their parents/caregivers to the PHCCs. Such children were assessed by the physicians at the health facilities. If the children were diagnosed with pneumonia, then they were advised to take oral antibiotics for 5 days at their home, based on the World Health Organization (WHO) criteria [27]. In the case of not having improvement with oral antibiotic therapy, the children were referred to the nearby health care facilities for appropriate injectable antibiotics [3]. Even from the beginning of the illness, if the children were suspected as a case of severe pneumonia at their respective homes by the trained health workers or if parents/caregivers observed the severity of illness including respiratory distress, then those children were referred and/or brought to the health care facilities (day-care clinics or hospitals). There they were evaluated by the study physician for enrolment into the study after fulfilling the eligibility criteria mentioned earlier. Written informed consent was obtained from the parents or legal guardians by the study nurses of all children before enrolment into the study. 

A questionnaire was used to collect demographic information, such as child’s age and sex, maternal and paternal age, education and occupation, number of family members, number of siblings, income and wealth, tertiles of the household (using indicators such as household materials for floor, roof, and walls, and household assets, including the type of latrine use and sources of drinking water, etc.), and clinical characteristics, including rapid breathing, lower chest wall in-drawing, hypoxemia, nasal discharge, and fever, and their disappearance in order to achieve the resolution of severe pneumonia. To ensure treatment at the day-care facilities, hospitals, and pediatric clinics in the vicinity were comparable to that provided at the study hospital, we trained the study physicians and nurses of day-care clinics, pediatric hospitals, and clinics and also equipped them with beds, oxygen, suction machines, nebulizers, and ensured the availability of antibiotics and supportive medicines. Children were admitted to the day-care clinics between 08:30 and 17:00 h on all seven days of the week, including on public holidays. The children admitted to the hospitals were managed following their respective protocols/guidelines. At the day-care centers, hospitals, and clinics, children were treated until resolution of clinical symptoms (no fast breathing, no chest wall in drawing, no nasal discharge, and no fever) and hypoxemia. After admission to the day-care facilities and hospitals, children were weighed by Seca weighing scale (Germany), and their recumbent length/standing height was measured, and they were placed on patients’ beds. Their parents or caregivers stayed with them. An electric suction machine was used to gently remove secretions from the throat and the nostrils. A pulse oximeter (OxiMax N-600, Nellcor, Boulder, CO, USA) was used to measure oxygen saturation, and if oxygen saturation was <90%, oxygen therapy was provided to them. Children having hypoxemia for more than six hours or at the closure of day-care (17:00 h) were referred to the secondary hospitals rather than sending them home. Children admitted to the day-care center were assessed before 17:00 h, and those in a stable condition and not requiring oxygen were sent home, advising their parents to bring their children back to the day-care center the next morning at 8:30 a.m. Children who were too ill to go home in the evening were referred to hospitals for treatment and care. Designated study staff accompanied all referral children to the hospitals. Children failing to present the following morning were visited at their homes by CHWs, to encourage their parents/caregivers to comply with the daily visit to receive treatment and care.

As in our study, we diagnosed severe pneumonia following evidence-based WHO guidelines that scrupulously adhered with the clinical presentation with or without chest X-ray; we did not evaluate the bacterial/viral etiology of pneumonia. For that, we empirically administered antibiotics irrespective of bacterial/viral species, which is also recommended by the WHO [28]. A recent pneumonia etiology research on children (PERCH) conducted in Asia and Sub-Sahara Africa revealed that the etiology of most of WHO classified childhood severe pneumonia was viral, followed by bacterial pathogens. Some of them had both co-infection of bacteria and viruses. The results underscore the importance of providing empirical antibiotics in these children, especially in resource-limited settings, such as Bangladesh [29]. Therefore, all the study children received a parenteral antibiotic named Ceftriaxone for ≥ 5 days, depending on their response. Breastfed children continued breastfeeding, and infant formula was given to non-breastfed infants 2 to 6 months of age, and those older than 6 months received milk-suji (a mixture of rice powder boiled in milk with sugar and vegetable oil, 67 kcal/100 g) [30]. All malnourished children received vitamin A, 100,000 IU for 6–12 months ages and 200,000 IU for older ages, a multivitamin (vitamin A palmitate 5000 IU, vitamin D 1000 IU, thiamine hydrochloride 1.6 mg, riboflavin 1 mg, pyridoxine hydrochloride 1 mg, nicotinamide 10 mg, calcium D-pantothenate 5 mg, and ascorbic acid 50 mg), 1 mL for 6–12 months ages and 2 mL for older children daily, folic acid 1.25 mg daily, elemental zinc 2 mg/kg daily, and potassium 4 mmol/ kg daily. The management continued until resolution of pneumonia, but for at least 5 days.

### 2.5. Sample Size Calculations

We assumed 10% exposure rates of each of the clinical indicators of severe pneumonia, i.e., fast breathing, lower chest wall in-drawing, and hypoxemia. At a 5% level of significance with 80% power, we estimated the sample size of 1276 children—638 each in the malnourished and well-nourished groups (case and control ratio of 1:1 and desired odds ratio was 2.0). We collected data from all the under-five children admitted to the study day-care and hospital facilities. 

### 2.6. Operational Definitions

Pneumonia is classified as severe pneumonia, pneumonia, or no pneumonia. Pneumonia is defined as fast breathing (≥ 50 breaths/min for 2–11 months of age, ≥ 40 breaths/min for 1–5 years of age), and/or chest in-drawing. Severe pneumonia is defined as the presence of cough or breathing difficulty with at least one of the following signs: central cyanosis, hypoxemia (oxygen saturation <90% on pulse oximetry), severe respiratory distress (grunting, very severe chest in-drawing), or signs of pneumonia with a general danger sign (inability to breastfeed or drink, lethargy, unconsciousness, and/or convulsions) [28]. Resolution/recovery means no rapid breathing, no lower chest wall in-drawing, no hypoxemia, no nasal discharge, and no fever. No rapid breathing is defined as respiratory rates of <50 breaths per minute for infant 2 to 12 months, and <40 breaths per minute for 12 to 59 months of age, no lower chest wall in-drawing means lower chest wall (lower ribs) does not go inward during inhalation, no hypoxemia is defined as oxygen saturation ≥ 90% with ambient air, and no fever means axillary temperature < 37.5 °C for ≥ 24 h [27].

### 2.7. Statistical Analysis

We analyzed the data by using SPSS for Windows (version 21.0; SPSS Inc., Chicago, IL, USA) and Epi Info (version 7.0, USD, Stone Mountain, GA, USA). On the basis of previous study findings [23,24,25], we assumed the clinical outcome of day-cares and hospitals would be similar, and therefore, we analyzed the data together to compare the malnourished and well-nourished children, and we did not compare the outcome of clinical features among day-cares versus hospitals in our study. As day-cares and hospitals were used for the same protocols for the management/treatment of children with severe pneumonia, we did not consider it as a variable, and for that reason, it was not analyzed. Basic demographic and clinical characteristics of categorical data were tabulated and analyzed as a proportion by Chi-square test, whereas normally distributed continuous variables were reported as mean (SD) by performing a Student’s *t*-test. The continuous variables data is not normally distributed, were analyzed by non-parametric (Mann–Whitney U) test (Table 1), and reported as a median (IQR). Differences in the proportions of the resolution of clinical features among the malnourished and well-nourished children on day six after admission in the health care facilities were compared by the Wald test (equality of two proportions test). A probability of less than 5% (*p* < 0.05) was considered statistically significant (Table 2). The strength of associations was determined by calculating the odds ratio (OR) and its 95% confidence intervals (CI). We used these statistics in our multiple logistic regressions (i.e., AOR) for all the possible confounding factors). In this study, we considered clinical characteristics (rapid breathing, lower chest wall in-drawing, hypoxemia, nasal discharge, and fever) as the dependent variables and all the demographic characteristics and children’s nutritional status as the independent variables (Table 3). We categorized the age of the children into <1 year (infant) and ≥ 1 year, as the infants had less immunity, hence were more vulnerable to developing pneumonia with complications [31]. We assessed wealth based on the possession of selected household assets and divided the participants into poor, middle, and rich tertiles [32]. The wealth tertiles were calculated based on the asset-related dichotomous variables (having or not having an asset) in the households. We used principal component analysis and produced a common factor score for each household. Then we divided the principal component score into tertiles to create three categories, where the first category represents poor households, and the third category represents rich households. 

## 3. Results

Of the total 2842 screened children, 1693 (59.6%) children with severe pneumonia were recruited, of which 1585 (93.6%) completed the study (Figure 1). Among the 1693 children with severe pneumonia, 443 (26.2%) had malnutrition, and 1248 (73.7%) were without malnutrition (the case-to-control ratio was 2.82) at baseline after recruitment. Two participants (0.1%) were excluded from the analysis due to missing data. The mean ± SD age of the children less than 12 months was 6.04 ± 1.46, and children of ≥ 12 months of age was 22.32 ± 9.58 months (Table 1). Among the participants, the number of well-nourished children was significantly greater (*p* < 0.001) compared to that of malnourished in both age groups (less than 12 and ≥ 12 months of age). Sixty-three percent of the total children were male (malnourished 65.7% and not malnourished 62.1%). The mean ± SD of maternal and paternal ages was 24.7 ± 5.0 and 31.9 ± 6.4 years. Maternal and paternal illiteracy was significantly higher among the malnourished children compared to those without malnutrition (18.3% vs. 8.7%; *p* < 0.001 and 20.7% vs. 12.9%; *p* < 0.001, respectively). Most of the mothers were housewives (87.6%), and the frequency for having a housewife mother was higher in well-nourished children than the malnourished children (89.0% vs. 83.8%; *p* = 0.004). Skilled workers, office non-executive, and executive fathers had significantly more well-nourished children compared to malnourished children (28.5% vs. 22.8%; *p* = 0.006). On the other hand, fathers who were day laborers or garments and industry workers had significantly more malnourished children compared to well-nourished children (23.2% vs. 17.9%; *p* = 0.006). The number of family members <5 was 51.0%, and ≥ 5 was 49.0%, and 39.7% had no or one sibling, whereas 39.3% and 21.0% had two and three or more siblings, respectively (Table 1). The median total household income of the malnourished children was USD 188 (IQR: 141, 270) and that for those without malnutrition USD 235 (IQR: 165, 253) (*p* < 0.001) (Table 1). Children of 41.5% and 30.5%, with and without malnutrition were categorized as poor, 33.9% and 35.9% as middle and 24.6% and 33.7% were in the rich wealth tertile, respectively, and those differences were significant (*p* < 0.001) (Table 1). After recruitment, the proportion of all children with fast breathing, lower chest wall in-drawing, hypoxemia, nasal discharge, and fever were 99.4%, 97.5%, 10.6%, 86.8%, and 84.8%, respectively; and they were not significantly different between the malnourished and the well-nourished children (Table 1). 

The basic demographic and clinical characteristics of the study children are shown in Table 1. 

Wald test analyses suggested significant differences of resolution of rapid breathing (*p* = 0.035), lower chest wall in-drawing (*p* = 0.019), and fever (*p* = 0.021) with well-nourished compared to malnourished children (Table 2). On day six of the study, the proportion of children with the resolution of fast breathing (86.5% vs. 90.2%), lower chest wall in-drawing (90.5% vs. 93.9%), and fever (92.2% vs. 95.2%) was significantly higher in the well-nourished compared to malnourished children (Table 2).

In the multivariable logistic regression model, children with an age of less than 1 year had significantly more rapid breathing (AOR 0.623, 95% CI 0.438–0.887, *p* = 0.009) and nasal discharge (AOR 0.695, 95% CI 0.546–0.884, *p* = 0.003) compared to those who were ≥ 1 year-old-children. Female children suffered more nasal discharge (AOR 1.295, 95% CI 1.027–1.633, *p* = 0.029) compared to males. Children with mothers with primary education (AOR 0.471, 95% CI 0.226–0.980, *p* = 0.044) recovered significantly more from fever compared to those whose mothers were illiterate. Malnourished children having severe pneumonia suffered more from rapid breathing (AOR 1.636, 95% CI 1.150–2.328 *p* = 0.006), chest wall in-drawing (AOR 1.698, 95% CI 1.113–2.590, *p* = 0.014), and fever (AOR 1.700, 95% CI 1.066–2.710, *p* = 0.026) compared to well-nourished children. Paternal occupation, father’s education, household members, number of siblings, and wealth tertiles were not significantly associated with clinical characteristics of the children. In these multiple models, the variance inflation factor was <1.5, which indicated the absence of multicollinearity among the explanatory variables (Table 3). 

## 4. Discussion

We compared the resolution of clinical features and baseline characteristics of severe pneumonia in malnourished children compared with their counterparts. We also explored the factors associated with the resolution of clinical characteristics (rapid breathing, lower chest wall in-drawing, hypoxemia, nasal discharge, and fever) on day six after admission to the health care facilities. We observed the outcomes of the malnourished children to be inferior compared to those without malnutrition. Children less than 1 year of age had more rapid breathing and nasal discharge compared to ≥ 1-year-old children, and the proportion of malnourished females and males was 1:1.3. Our finding is in discordance with another study conducted in urban western Maharashtra, India, that reported a slightly higher proportion of female malnourished (female:male = 1:0.9) children [33]. 

Female children more often had suffered from nasal discharge compared to males. Expectedly, mothers and fathers of the malnourished children were less educated than the well-nourished children; the proportion of malnourished children was higher among the less educated mothers and fathers. Gradually increasing years of maternal and paternal education were associated with reducing the proportion of malnourished children. One study in Bangladesh showed that parents’ level of education and poor household income were the main risk factors for childhood stunting and being underweight [12]. This is in corroboration with the Uganda Demographic and Health Survey (UDHS) data that observed that mothers with primary and secondary levels of education had lower odds of having stunted (OR 0.78; Cl 0.62–0.97 and OR 0.64; Cl 0.47–0.88, respectively) and underweight children (OR 0.52; Cl 0.31–0.84) compared to those with no formal education [34]. Children of mothers having primary education recovered significantly more from fever compared to those whose mothers were illiterate. Most (87.7%) of the mothers in our study were housewives, who had a lesser proportion of malnourished children compared to the mothers engaged in various formal and informal occupations. Studies in Uganda and Mozambique observed a lesser proportion of malnourished children among mothers in professional/formal occupations compared to mothers in other occupations [35], including agricultural and manual work [34]. Mothers engaged in agricultural and other manual works had less time for providing care to their children than housewives [36]. Regular higher income of the mothers informal/professional occupation might have contributed to better food for their children [37]. 

Less than half of the fathers were involved in their own business, rickshaw pullers, push cart pullers, taxi, bus, and tempo drivers among both the malnourished and well-nourished children. Malnourished children compared to well-nourished children more often had day laborer, garment and industry worker fathers. In contrast, well-nourished children more often had fathers who were skilled workers, non-executive, and executive officers. A study in urban Nigeria found that childhood malnutrition was associated with father’s involvement in semi-skilled jobs, such as public minibus drivers, unskilled labor, such as farming or petty trading, and unemployed [38]. We have a similar finding in our study—the median household income of the well-nourished children was higher (USD 235) than the malnourished children (USD 188). The national average monthly household income was USD 188, and for the urban area, it was 266 in 2016 [39]. The maximum monthly household income was USD 270, and the minimum was USD 141 among the malnourished, and USD 353 and USD 165, respectively, among the well-nourished children, reflecting a huge disparity in the participant’s community. Studies in Ghana reported similar findings—the family income of 36.8% of malnourished children was ≤ USD 100 compared to 12.2% of the well-nourished children [40], and family income was reversely related to malnutrition (7.1% vs. 21.2% income > USD 171 malnourished and well-nourished children, respectively) [41]. In our study, a higher proportion of families of the malnourished children was in the poor wealth tertile than the families of the well-nourished children (41.5% vs. 30.5%); a similarly higher proportion of well-nourished children’s family was in the richer tertile than the family of the malnourished children (33.7 vs. 24.6). This finding is consistent with an earlier study’s findings [40].

A number of studies revealed that malnourished children had less age-specific fast breathing and lower chest wall in-drawing compared to well-nourished children due to poor inflammatory response [42], poor muscle mass [43], and hypokalemia [44] in malnourished children [45]. However, in our study, the children who had suffered from rapid breathing and lower chest wall in-drawing were comparable in both groups. Well-nourished children suffered more from hypoxemia, nasal discharge, and fever. One review article described that in cases of infection, clinical manifestation as fever is not apparent; therefore, infection among malnourished children might be difficult to diagnose [46]. Our study revealed that malnourished children suffered more from rapid breathing, lower chest wall in-drawing, and fever compared to well-nourished children. Earlier studies showed that the immune response of malnourished children is inadequate, and the observed clinical features may be explained by that observation; thus, they need greater care with more vigilance than the well-nourished [46,47]. However, in our study, we did not observe the same findings. 

### Limitations and Strengths of This Study

There are some limitations to our study. First, we excluded all the severely malnourished children from this study. Thus, their inclusion could have resulted in different outcomes. Second, 40.4% of children were excluded after screening and after enrollment, and an additional 6.4% did not complete the study. These exclusions and dropouts might have influenced our study findings. Third, as we did not perform blood and nasal swab/sputum culture for the diagnosis of bacterial/viral species, there might be different symptoms and clinical outcomes between bacterial and viral infection among the malnourished and well-nourished children. 

Despite the limitations, our study findings provide some insights for future modification/separate clinical guidelines for the management of malnourished under-five children suffering from severe pneumonia, the top-most cause of under-five death worldwide. This might ultimately help in preventing childhood morbidity and mortality in order to achieve the SDGs.

## Figures and Tables

**Figure 1 children-08-00778-f001:**
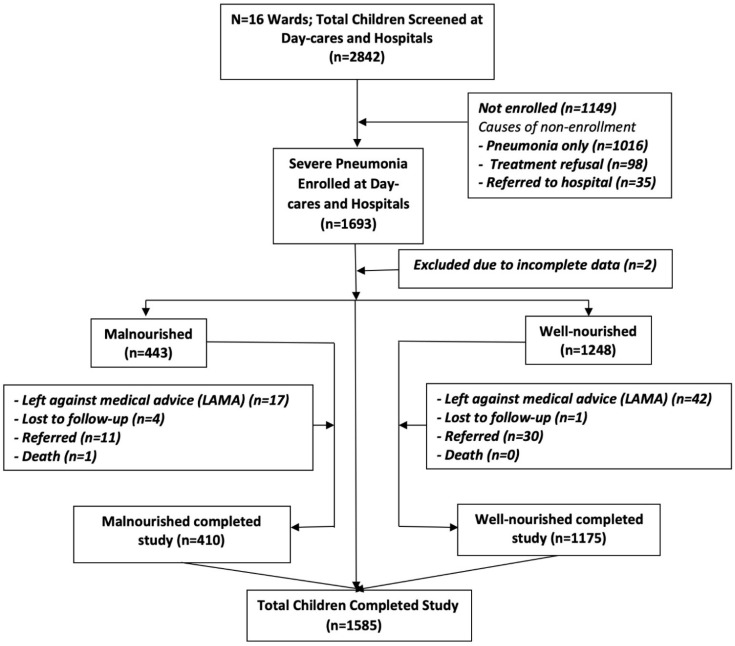
Flow chart of study participants.

**Table 1 children-08-00778-t001:** Basic Demographic and Clinical Characteristics of the Study Children at Admission.

Characteristic	Total (*n* = 1693) *n* (%)	Malnourished (*n* = 443) *n* (%)	Well-Nourished (*n* = 1248) *n* (%)	*p*-Value
Demographic characteristics				
Child’s age in months				
Less than 12 months (Mean ± SD) 6.04 ± 1.46	1039 (61.4)	221 (21.3)	817 (78.7)	<0.001
≥12 months (Mean ± SD) 22.32 ± 9.58	654 (38.6)	222 (34.0)	431 (66.0)	
Sex of the children (male)	1066 (63.0)	291 (65.7)	775 (62.1)	0.215
Maternal age in years, Mean (SD)	24.7 (5.0)	25.0 (5.3)	24.6 (4.9)	0.254
Paternal age in years, Mean (SD)	31.9 (6.4)	32.1 (6.9)	31.8 (6.2)	0.351
Maternal education (years of schooling)				
Illiterate (0 year); n (%)	190 (11.2)	81 (18.3)	109 (8.7)	<0.001
Primary (1–5 years); n (%)	522 (30.8)	149 (33.6)	373 (29.9)	
Secondary (6–10 years); n (%)	773 (45.7)	181 (40.9)	591 (47.4)	
Higher (≥11 years); n (%)	207 (12.2)	32 (7.2)	174 (14.0)	
Paternal education (years of schooling)				
Illiterate (0 year); n (%)	252 (14.9)	91 (20.7)	161 (12.9)	<0.001
Primary (1–5 years); n (%)	483 (28.5)	143 (32.6)	340 (27.3)	
Secondary (6–10 years); n (%)	677 (40.0)	160 (36.5)	516 (41.5)	
Higher (≥11 years); n (%)	274 (16.2)	45 (10.3)	228 (18.3)	
Maternal occupation				
Housewife	1483 (87.6)	371 (83.8)	1110 (89.0)	0.004
Other (formal and informal)	209 (12.3)	72 (16.3)	137 (11.0)	
Paternal occupation				
Day labor, Garment/Industry worker	326 (19.3)	102 (23.2)	223 (17.9)	0.006
Skilled worker, Office Non-executive, Office executive	455 (26.9)	100 (22.8)	355 (28.5)	
Business/Rickshaw/Push cart puller/Taxi/Bus/Tempo driver	735 (43.4)	194 (44.2)	541 (43.4)	
Other (informal)	169 (10.0)	42 (9.6)	126 (10.1)	
Number of family/household members				
<5 members; n (%)	863 (51.0)	243 (54.9)	620 (49.7)	0.063
≥5 members; n (%)	829 (49.0)	200 (45.2)	627 (50.3)	
Number of siblings				
One or no; n (%)	672 (39.7)	169 (38.2)	502 (40.3)	0.586
Two; n (%)	665 (39.3)	174 (39.3)	490 (39.3)	
Three or more; n (%)	355 (21.0)	100 (22.6)	255 (20.5)	
Household income (USD/month), Median (IQR)	USD 223 (165, 353)	USD 188 (141, 270)	USD 235 (165, 353)	<0.001 ^¶^
Wealth tertiles				
Poor; n (%)	565 (33.4)	184 (41.5)	380 (30.5)	<0.001
Middle; n (%)	597 (35.8)	150 (33.9)	447 (35.9)	
Rich; n (%)	530 (31.3)	109 (24.6)	420 (33.7)	
Presence of clinical characteristics at recruitment in the health care facilities				
Rapid breathing; n (%)	1682 (99.4)	440 (99.3)	1240 (99.4)	0.935
Lower chest wall in drawing; n (%)	1651 (97.5)	432 (97.5)	1217 (97.5)	0.999
Hypoxemia (SPO_2_ < 90%); n (%)	179 (10.6)	39 (9.1)	140 (11.6)	0.168
Nasal discharge; n (%)	1470 (86.8)	377 (85.1)	1091 (87.4)	0.215
Fever; n (%)	1436 (84.8)	371 (83.8)	1064 (85.3)	0.446

Chi-square test, Students *t*-test, ^¶^ Mann–Whitney U test. IQR = Interquartile range, SD = Standard Deviation. *p*-Values indicate statistical significance at α = 0.05.

**Table 2 children-08-00778-t002:** The resolution of clinical features of severe pneumonia in malnourished children compared to well-nourished children on day six after admission.

Resolution on Day Six after Admission into the Health Care Facilities	Malnourished Children n (%)	Well-Nourished Children n (%)	*p*-Value
No rapid breathing	365 (86.5) N = 422	1077 (90.2) N = 1194	0.035
No lower chest wall in-drawing	382 (90.5) N = 422	1121 (93.9) N = 1194	0.019
No hypoxemia	404 (99.5) N = 406	1162 (99.5) N = 1168	1.000
No nasal discharge	319 (75.6) N = 422	874 (73.2) N = 1194	0.335
No fever	389 (92.2) N = 422	1137 (95.2) N = 1194	0.021

*p*-Values indicate statistical significance at α = 0.05.

**Table 3 children-08-00778-t003:** Multivariable logistic regression model showed the association between independent (x) background factors and dependent (y) variables as different clinical characteristics.

Variables	Rapid Breathing (R² = 0.039)		Chest Wall in Drawing (R² = 0.031)		Hypoxemia (R² = 0.220)		Nasal Discharge (R² = 0.025)		Fever (R² = 0.040)	
	AOR (95% CI)	*p*-Value	AOR (95% CI)	*p*-Value	AOR (95% CI)	*p*-Value	AOR (95% CI)	*p*-Value	AOR (95% CI)	*p*-Value
Age										
<1 year	ref.		ref.		ref.		ref.		ref.	
≥1 year	0.623 (0.438–0.887)	0.009	0.766 (0.504–1.165)	0.213	0.365 (0.063–2.106)	0.260	0.695 (0.546–0.884)	0.003	0.692 (0.432–1.109)	0.126
Sex										
Male	ref.		ref.		ref.		ref.		ref.	
Female	1.081 (0.775–1.506)	0.648	0.935 (0.621–1.407)	0.747	1.877 (0.442–7.977)	0.393	1.295 (1.027–1.633)	0.029	0.896 (0.565–1.420)	0.639
Father’s Education										
Illiterate	ref.		ref.		ref.		ref.		ref.	
Primary	1.022 (0.583–1.794)	0.939	1.011 (0.525–1.947)	0.973	0.000 (0.000–0.000)	0.992	0.932 (0.633–1.375)	0.724	1.612 (0.774–3.359)	0.202
Secondary	1.361 (0.778–2.379)	0.280	1.278 (0.664–2.461)	0.463	0.455 (0.065–3.189)	0.428	1.233 (0.837–1.816)	0.290	1.007 (0.460–2.204)	0.986
Higher	1.862 (0.918–3.775)	0.085	1.928 (0.832–4.469)	0.126	1.424 (0.142–14.331)	0.764	1.168 (0.706–1.932)	0.546	0.870 (0.309–2.448)	0.792
Father’s Occupation										
Day labor, Garment/Industry worker	ref.		ref.		ref.		ref.		ref.	
Skilled worker, Office non-executive & executive	0.918 (0.547–1.542)	0.747	0.987 (0.517–1.886)	0.969	1.384 (0.102–18.812)	0.807	1.092 (0.764–1.562)	0.629	1.019 (0.496–2.095)	0.959
Business/Rickshaw/Push cart puller/Taxi/Bus/Tempo driver	1.010 (0.639–1.597)	0.966	1.241 (0.709–2.172)	0.449	2.009 (0.217–18.615)	0.539	0.969 (0.703–1.338)	0.850	1.309 (0.701–2.443)	0.398
Other (informal)	1.339 (0.730–2.456)	0.345	1.648 (0.793–3.424)	0.181	0.000 (0.000–0.000)	0.995	1.060 (0.675–1.664)	0.800	1.346 (0.565–3.208)	0.502
Mother’s Education										
Illiterate	ref.		ref.		ref.		ref.		ref.	
Primary	0.568 (0.320–1.009)	0.054	0.790 (0.399–1.565)	0.499	1.088 (0.081–14.646)	0.949	1.078 (0.702–1.654)	0.732	0.471 (0.226–0.980)	0.044
Secondary	0.761 (0.432–1.340)	0.344	0.880 (0.441–1.755)	0.716	2.836 (0.193–41.778)	0.448	1.161 (0.752–1.794)	0.500	0.664 (0.327–1.350)	0.258
Higher	0.738 (0.342–1.592)	0.438	0.602 (0.225–1.609)	0.311	0.000 (0.000–0.000)	0.995	1.177 (0.664–2.084)	0.577	0.862 (0.312–2.384)	0.775
Mother’s Occupation										
Housewife	ref.		ref.		ref.		ref.		ref.	
Others	0.641 (0.354–1.158)	0.14	0.686 (0.342–1.377)	0.289	2.265 (0.377–13.603)	0.371	1.148 (0.804–1.639)	0.447	1.135 (0.571–2.256)	0.718
Household Members										
<5 members	ref.		ref.		ref.		ref.		ref.	
≥5 members	0.980 (0.674–1.427)	0.918	1.494 (0.953–2.343)	0.080	0.593 (0.060–5.883)	0.655	0.984 (0.756–1.281)	0.906	0.693 (0.409–1.172)	0.171
Number of Siblings										
One or no	ref.		ref.		ref.		ref.		ref.	
Two	1.111 (0.774–1.595)	0.568	0.931 (0.595–1.456)	0.753	0.936 (0.128–6.830)	0.948	1.174 (0.910–1.514)	0.218	1.066 (0.659–1.726)	0.793
Three or more	1.064 (0.636–1.780)	0.814	0.882 (0.489–1.591)	0.677	4.996 (0.412–60.646)	0.207	1.064 (0.742–1.526)	0.736	0.912 (0.431–1.933)	0.811
Wealth Tertiles										
Poor	ref.		ref.		ref.		ref.		ref.	
Middle	1.143 (0.765–1.710)	0.514	0.829 (0.514–1.337)	0.443	0.217 (0.022–2.184)	0.195	1.002 (0.748–1.341)	0.990	1.647 (0.937–2.894)	0.083
Rich	0.767 (0.482–1.221)	0.264	0.609 (0.350–1.057)	0.078	0.730 (0.118–4.500)	0.734	1.205 (0.880–1.649)	0.244	1.505 (0.801–2.826)	0.204
Nutrition Status										
Well-nourished	ref.		ref.		ref.		ref.		ref.	
Malnourished	1.636 (1.150–2.328)	0.006	1.698 (1.113–2.590)	0.014	1.098 (0.195–6.167)	0.916	1.012 (0.776–1.320)	0.932	1.700 (1.066–2.710)	0.026

AOR = Adjusted Odds Ratio, Cl = Confidence Interval. *p*-Values indicate statistical significance at α = 0.05.

## Data Availability

The data presented in this study are available on request from the corresponding author. The data are not publicly available due to research participants’ privacy.

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
