# Peer review of "Clinical Characteristics and Underlying Factors Related to Recovery from Severe Pneumonia in Under-Five Children with or without Malnutrition Treated at Health Care Facilities in Bangladesh"

_children, 2021, doi:10.3390/children8090778_

Round 1

Reviewer 1 Report

The authors present a study on the impact of malnourishment on pneumonia outcomes. Several concerns are apparent that limit the impact of the study. 

1) The authors need to explain the criteria used to determine a diagnosis of pneumonia. While there is a description of severe pneumonia, it is not clear what the original operational definition of pneumonia was to get into the study and that is key to understanding the results. 

2) The flow diagram for participants notes randomization, but the methods do not describe when this happens. It is not clear if the participants were randomized to daycare or hospital care. 

3) The study appears to divide participants by care in day care versus hospital care, but none of the analyses are presented through this lens so it is unclear how or if this factored into outcomes. 

Author Response

Date:   26 August, 2021

Subject: Re-submission of the [Children] Manuscript ID: children-1283037

Thank you very much again for your valuable comments and further consideration of my manuscript for possible acceptance.

To improve the English revision, we all the authors worked on it and also checked by a native English language person from Hiroshima University Writing Center, 1-2-2 Kagamiyama, Higashi-Hiroshima City Hiroshima, Japan 739-8512, TEL: +81-82-424-6201.

Please find below the point-by-point responses and all the changes are marked by “Track Changes” within the revised manuscript.

Best regards,

KATM Ehsanul Huq
Doctoral student
Health Sciences Major
Graduate School of Biomedical & Health Sciences
Hiroshima University, Japan
Mobile:080 6266 8578

Review Report Form 1:

Comment 1) The authors need to explain the criteria used to determine a diagnosis of pneumonia. While there is a description of severe pneumonia, it is not clear what the original operational definition of pneumonia was to get into the study and that is key to understanding the results. 

Response: Thank you very much for the insightful comment. On the basis of clinical presentation, pneumonia is classified as severe pneumonia, pneumonia, or no pneumonia. Pneumonia is defined as fast breathing (≥50 breaths/min for 2-11 months of age, ≥40 breaths/min for 1-5 years of age), and/or chest indrawing. Severe pneumonia is defined as present of cough or breathing difficulty with at least one of the following signs: central cyanosis, hypoxaemia (oxygen saturation < 90% on pulse oximetry), severe respiratory distress (grunting, very severe chest indrawing), or signs of pneumonia with a general danger sign (inability to breastfeed or drink, lethargy, unconsciousness and/or convulsions) We incorporated these sentences in the “Operational Definitions” of main text in page 5,    line 213-220.     

Comment 2) The flow diagram for participants notes randomization, but the methods do not describe when this happens. It is not clear if the participants were randomized to daycare or hospital care.

Response: Thank you very much for your excellent comment. To avoid the reader’s confusion, we revised the diagram and deleted the portion that was not described in the methodology. We screened a total of 2,842 children in the day-cares and hospitals and after checking the inclusion and exclusion criteria, 1,693 under five children with severe pneumonia were enrolled in day-cares and hospitals. We revised these changes in page 7 (Figure 1).         

Comment 3) The study appears to divide participants by care in day care versus hospital care, but none of the analyses are presented through this lens so it is unclear how or if this factored into outcomes. 

Response: Thanks for your core query. Actually, the outcome comparing the daycare vs. hospital care was a separate manuscript that is yet to be published. In our analysis, we combined the data of severe pneumonia children who were admitted in the day-cares and hospitals. Then, we divided the participants on the basis of nutritional status (malnutrition and well-nutrition). For that reason, we did not compare the outcome of clinical features among day-cares versus hospitals in our study. As day-cares and hospitals were used the same protocol for the management/treatment of children with severe pneumonia, we did not consider it as a variable and for that reason not analyzed. We revised these sentences in page 3, line 100-103, and incorporated the above statement in page 5, line 232-235.    

Reviewer 2 Report

One area of concern is that all children received antibiotics regardless of if it was bacterial or viral pneumonia. It is unclear if this intervention plays a role in outcome measurement given some patient received medication that clearly would have helped their diagnosis while some did not (the viral group). If there were an excess bacterial pneumonia proportion of patients in the non-malnurished group, the results may simply indicate that antibiotics are working, rather than a role of nutritional status.

In terms of the results, it seems that much of the data says similar things, namely that socioeconomic support results in less nutritional deficiency. It was nice the way it was clearly shown malnourished kids had worse outcomes, but I'm not sure if the objective of associating pneumonia outcomes specifically to nutritional status is clear given how multifactorial it is. This does lay out good description of raw data, but conclusions are hard to draw because etiology of each diagnosis is so unclear. Pneumonia is not a specific pathogen, but the specific etiology is important in understanding symptoms and outcomes. For example, viral pneumonia can explain different symptoms than bacterial. A better way to frame the paper, may be to state that kids who are malnourished and have pneumonia have worse outcomes, rather than focusing on the pneumonia itself correlating with specific symptoms and outcomes.

Author Response

Date:   26 August, 2021
Subject: Re-submission of the [Children] Manuscript ID: children-1283037

Thank you very much again for your valuable comments and further consideration of my manuscript for possible acceptance.

To improve the English revision, we all the authors worked on it and also checked by a native English language person from Hiroshima University Writing Center, 1-2-2 Kagamiyama, Higashi-Hiroshima City Hiroshima, Japan 739-8512, TEL: +81-82-424-6201.

Please find below the point-by-point responses and all the changes are marked by “Track Changes” within the revised manuscript.

Best regards,

KATM Ehsanul Huq
Doctoral student
Health Sciences Major
Graduate School of Biomedical & Health Sciences
Hiroshima University, Japan
Mobile:080 6266 8578

Review Report Form 2:

Comment 1) One area of concern is that all children received antibiotics regardless of if it was bacterial or viral pneumonia. It is unclear if this intervention plays a role in outcome measurement given some patient received medication that clearly would have helped their diagnosis while some did not (the viral group). If there were an excess bacterial pneumonia proportion of patients in the non-malnurished group, the results may simply indicate that antibiotics are working, rather than a role of nutritional status.

Response: We thank for your valuable comments. “As in our study we diagnosed severe pneumonia following evidence-based WHO guideline that scrupulously adhered with clinical presentation with or without chest X-ray, we did not evaluate the bacterial/viral etiology of pneumonia. For that we empirically administered antibiotics irrespective of bacterial/viral species that is also recommended by the WHO. It is important to note that a recent pneumonia etiology research on children (PERCH) conducted in Asia and Sub-Sahara Africa revealed that the etiology of most of WHO classified childhood severe pneumonia was viral followed by bacterial pathogens. Some of them had both co-infection of bacteria and virus. The results underscore the importance of providing empirical antibiotics in these children especially in resource limited settings like Bangladesh”. We added the above statements in page 4, line 184-193. We have also added a related limitation section “Third, as we did not perform blood and nasal swab/sputum culture for the diagnosis of bacterial/viral species, there might be different symptoms and clinical outcomes between bacterial and viral infection among the malnourished and well-nourished children.” We added the above sentences in the ‘Limitations and Strengths of This Study’ of Discussion page 2, lines 76-79.     

Comment 2) In terms of the results, it seems that much of the data says similar things, namely that socioeconomic support results in less nutritional deficiency. It was nice the way it was clearly shown malnourished kids had worse outcomes, but I'm not sure if the objective of associating pneumonia outcomes specifically to nutritional status is clear given how multifactorial it is. This does lay out good description of raw data, but conclusions are hard to draw because etiology of each diagnosis is so unclear. Pneumonia is not a specific pathogen, but the specific etiology is important in understanding symptoms and outcomes. For example, viral pneumonia can explain different symptoms than bacterial. A better way to frame the paper, may be to state that kids who are malnourished and have pneumonia have worse outcomes, rather than focusing on the pneumonia itself correlating with specific symptoms and outcomes.

Response: We quite concur with your thoughtful suggestions. We also agree that there might be different symptoms and clinical outcomes between bacterial and viral infection among the malnourished and well-nourished children. However, this study followed WHO recommended diagnosis and treatment of pneumonia on the basis of the clinical spectrum that did not allow us to perform blood and nasal swab/sputum culture for the diagnosis of bacterial/viral species.   
